# Preformulation Study of Electrospun Haemanthamine-Loaded Amphiphilic Nanofibers Intended for a Solid Template for Self-Assembled Liposomes

**DOI:** 10.3390/pharmaceutics11100499

**Published:** 2019-09-29

**Authors:** Khan Viet Nguyen, Ivo Laidmäe, Karin Kogermann, Andres Lust, Andres Meos, Duc Viet Ho, Ain Raal, Jyrki Heinämäki, Hoai Thi Nguyen

**Affiliations:** 1Faculty of Pharmacy, Hue University of Medicine and Pharmacy, Hue University, 06 Ngo Quyen, Hue City 530000, Viet Nam; nvietkhan@gmail.com (K.V.N.); hovietduc661985@gmail.com (D.V.H.); hoai77@gmail.com (H.T.N.); 2Institute of Pharmacy, Faculty of Medicine, University of Tartu, Nooruse str. 1, 54011 Tartu, Estonia; ivo.laidmae@ut.ee (I.L.); kkogermann@gmail.com (K.K.); andres.lust@ut.ee (A.L.); andres.meos@ut.ee (A.M.); ain.raal@ut.ee (A.R.)

**Keywords:** haemanthamine, plant-origin alkaloid, electrospinning, amphiphilic nanofibers, self-assembled liposomes, physical solid-state properties, drug release

## Abstract

Haemanthamine (HAE) has been proven as a potential anticancer agent. However, the therapeutic use of this plant-origin alkaloid to date is limited due to the chemical instability and poorly water-soluble characteristics of the agent. To overcome these challenges, we developed novel amphiphilic electrospun nanofibers (NFs) loaded with HAE, phosphatidylcholine (PC) and polyvinylpyrrolidone (PVP), and intended for a stabilizing platform (template) of self-assembled liposomes of the active agent. The NFs were fabricated with a solvent-based electrospinning method. The chemical structure of HAE and the geometric properties, molecular interactions and physical solid-state properties of the NFs were investigated using nuclear magnetic resonance (NMR) spectroscopy, scanning electron microscopy (SEM), photon correlation spectroscopy (PCS), Fourier transform infrared (FTIR) spectroscopy, X-ray powder diffraction (XRPD) and differential scanning calorimetry (DSC), respectively. An in-house dialysis-based dissolution method was used to investigate the drug release in vitro. The HAE-loaded fibers showed a nanoscale size ranging from 197 nm to 534 nm. The liposomes with a diameter between 63 nm and 401 nm were spontaneously formed as the NFs were exposed to water. HAE dispersed inside liposomes showed a tri-modal dissolution behavior. In conclusion, the present amphiphilic NFs loaded with HAE are an alternative approach for the formulation of a liposomal drug delivery system and stabilization of the liposomes of the present alkaloid.

## 1. Introduction

Today, nanoscale drug delivery systems (DDSs) provide a novel alternative for conventional therapeutic approaches for potential new drugs of plant origin, or for established synthetized drugs which have been forgotten due to their challenging properties. Compared to the conventional dosage forms, nanotechnology-based DDSs enable the enhancement of biological activity, reduce therapeutic dose, reduce side-effects, deliver active ingredients to the desired target and improve physicochemical stability. The common problems associated with the potential active agents of plant origin, e.g., poor water-solubility, physicochemical stability, permeability, safety and efficiency of challenging drug molecules, can be solved when these are formulated into nanoscale DDSs, such as liposomes, nanoparticles, dendrimers, nanocapsules, ethosomes and polymersomes [1]. 

Haemanthamine (HAE; Figure 1) is a crinine-type alkaloid isolated from the plant of the family Amaryllidaceae [2]. HAE is reported to have many prominent bioactivities (potential therapeutic effects), including neuromuscular transmission, antimalarial, antioxidant, anticonvulsant, butyrylcholinesterase-inhibitory activity, antiviral and anticancer activity [2,3]. Numerous studies have shown that HAE has a strong cytotoxic activity being effective against human melanoma (SK-MEL-28), human lung carcinoma (A549), human T lymphoblast leukemia (MOLT-4), human esophageal squamous carcinoma (OE21), mouse melanoma (B16-F10), human hepatocellular carcinoma (HepG2), human brain glioma (Hs683), human breast adenocarcinoma (MCF-7) and human acute T cell leukemia (Jurkat) [2,3,4,5]. Havelek et al. [2] reported that the treatment of accumulating cells at G1 and G2 stages suppressed cell viability and mitochondrial membrane potential, increased apoptosis through cell cycle progressions and slowed down proliferation. According to Pellegrino et al. [6], HAE suppressed the growth of cancer cells by binding at the A-site of the large ribosomal subunit and, consequently, activating a p53-dependent antitumoral nucleolar stress response [6].

From the pharmaceutical formulation point of view, HAE is a very challenging plant-origin active agent since it is a large molecule agent, chemically instable (acid-labile) and poorly soluble in water. We believe that this a major reason for the absence of the relevant formulation and clinical studies in the current literature associated with the present potentially anti-cancer drug. Our scientific hypothesis is that the development of an advanced nanoscale DDS for HAE would enhance the physicochemical stability and drug delivery of the present anticancer drug candidate, and ultimately enable effective therapy.

The drug-loaded liposomes made of lipid bilayer vesicles have been widely used as nanoscale DDSs for, e.g., combating multi-drug resistance, targeting anticancer drug, vaccine and protein delivery, developing long-term circulating DDSs and enhancing gene therapy [7]. However, the well-known limitation associated with the formulation and use of pharmaceutical liposomes is their limited physical stability under storage. Yu and coworkers [8] introduced amphiphilic electrospun nanofibers (NFs) intended for a stabilizing platform of the self-assembled liposomes. The NFs were composed of hydrophilic polyvinylpyrrolidone (PVP) and phosphatidylcholine (PC) (Figure 1), and the corresponding self-assembled PC liposomes were spontaneously formed as the NFs were exposed to water. The present approach has major advantages over the conventional fabrication methods of liposomes since no heating, cooling, agitating, sonicating or sterilizing phases are needed in a fabrication process, and the liposomes can be stored as an “inactive” solid form in the nanofibrous platform prior to use [8]. The study published by Yu et al. [8] inspired us to make attempts to develop such stabilized nanoscale DDSs also for HAE. Our study was conducted to further develop the pioneer work of Yu and co-workers, and to extend the use of such amphiphilic nanofibers in the field of Pharmacy. To date, such amphiphilic nanofibers have not been used as DDSs and as a formulation strategy for active pharmaceutical ingredients. The loading of a high-molecule active agent in such nanofibers to maintain the formation of self-assembled liposomes is a paramount challenge.

In the recent years, the interest in the development of nanofibrous DDSs for therapeutic small molecules and large molecules (biologicals) has been greatly increased [9]. Electrospinning (ES) is the technique of choice for preparing composite nanomaterials and nanofibrous DDSs. ES is a rapid, continuous and low-cost fabrication method that can be readily scaled-up to large-scale production. The active agents (including the challenging plant-origin molecular solids) can be loaded in the electrospun NFs resulting in the desirable properties of the NFs, including a thin diameter, uniform structure, high surface area and porosity [9,10,11]. In spite of well-known advantages, however, the use of ES and its modifications (coaxial and melt ES) in the pharmaceutical formulation development is still rather limited.

The main objectives of the present study were (1) to formulate novel amphiphilic electrospun NFs for HAE, and to use such nanofibrous platforms as a solid template for self-assembled liposomes of HAE, and (2) to investigate the formation, drug-polymer interactions and physical solid-state properties of the abovementioned nanofibrous templates and self-assembled liposomes. The present novel nanoformulation could be applicable for anticancer drug therapy and for administering via a parenteral route. The nanoscale solid template and self-assembled liposomes could provide an alternative formulation strategy for HAE, thus enabling to enhance the chemical stability and antitumor efficacy of this plant-origin alkaloid. 

## 2. Materials and Methods 

### 2.1. Materials

HAE (a white, crystalline powder, purity NLT 95%) was isolated from *Zephyranthes ajax* Hort. belonging to the family Amaryllidaceae in the Hue University of Medicine and Pharmacy, Hue City, Vietnam. PVP (Kollidon 90F K90) was obtained from BASF SE, Germany. Soybean PC (Lipoid S-100) was obtained from Lipoid GmbH, Ludwigshafen, Germany. Ethanol, methanol and acetonitrile were purchased from Merck GmbH, Germany. Purified water (HPLC grade) was used as an aqueous media. Sodium dodecylsulphate, sodium phosphate and ammonium acetate were purchased from Sigma-Aldrich C.C., St. Louis, MO, USA. 

### 2.2. NMR Structure Analysis 

Nuclear magnetic resonance (NMR) spectra of HAE were obtained with a Bruker Avance 500 NMR spectrometer (^1^H NMR for 500 MHz, ^13^C NMR for 125 MHz) (Bruker BioSpin Corporation, Billerica, MA, USA), with tetramethylsilane (TMS) as an internal reference.

### 2.3. Fabrication of Amphiphilic Nanofibers

For preparing the solutions for ES, soybean PC (0.15 g) was first dissolved in ethanol and stirred in a magnetic stirrer for one hour. Subsequently, PVP (0.30 g) was added in the solution and stirred for at least 17 h in a magnetic stirrer at an ambient room temperature (22 ± 2 °C). Finally, HAE (21.0 mg) was added and stirred for 6 h to form the solution for fabricating the amphiphilic NFs. The present solution consisted of soybean PC and PVP at a weight ratio of 1:2. In addition, we tested the ES of amphiphilic NFs with the solution consisting of soybean PC and PVP at a weight ratio of 1:4. 

The amphiphilic NFs intended for a solid template for self-assembled liposomes were fabricated with an ESR200RD robotized ES and electrospraying system (NanoNC Co. Ltd., Seoul, Korea). The robotized ES system was composed of a programmable syringe pump, a special 2.5 mL syringe (spinneret) equipped with a metal 25G needle, a high-voltage power supply (Model HV30) and a collector plate covered with aluminum foil (Figure 2). The flow rate of the solution was set at 1.0 mL/h, and the operating voltage between the spinneret and the grounded collector plate was set at 10 kV. The distance between the needle tip and the collector plate was approximately 12 cm. The ES experiments were carried out at an ambient room temperature (22 ± 2 °C) and a relative humidity (RH) of 18–20%. The fiber samples were stored in a zip-lock plastic bag in a refrigerator (3–8 °C) prior to analysis. 

### 2.4. The Geometric Properties and Surface Morphology of NFs 

A digital microscope (CETI, Medline Scientific Limited, Chalgrove Oxon, UK) and a high-resolution SEM (NanoSEM 450, FEI Corp., Hillsboro, OR, USA) were used for investigating the physical appearance, geometric properties (diameter and shape) and surface morphology of the nanofibrous templates, respectively. The SEM experiments were conducted as described in our previous study, with slight modifications [12,13]. For SEM, the NF samples were attached onto aluminum stubs and coated with a 3 nm gold layer with a magnetron sputter. The SEM diameter of NFs was measured with ImageJ 1.50b software (National Institutes of Health, Rockville Pike, Bethesda, MD, USA). The measurements were performed for at least 100 individual NFs (*n* = 100). 

### 2.5. Optical Microscopy of Self-Assembled Liposomes

For assessing the spontaneous formation of self-assembled liposomes, 1.0 mL of distilled water was added onto the pre-weighted sample (100 mg) of HAE-loaded NFs. The mixture was then gently manually shaken for about 1–2 min at an ambient room temperature (22 ± 2 °C) to obtain a homogenous dispersion. The dispersion was equilibrated for at least 10 min for forming the liposomes by self-deposition. The sample dilution was required for acquiring proper images. Then, the formation of the self-assembled liposomes was verified and imaged with an optical microscope CETI MAGTEX (Medline Scientific Limited, Chalgrove Oxon, UK).

### 2.6. Photon Correlation Spectroscopy (PCS) 

Photon correlation spectroscopy (PCS instrument Nicomp submicron particle analyzer model 380, Nicomp Inst Corp, Santa Barbara, CA, USA) was used for measuring the size and size distribution of liposomes. The nanofibrous template was first wetted by adding 5.0 mL of distilled water onto the pre-weighted sample (100 mg) of NFs (as optimized earlier in our laboratory). After hydration of NFs, the dispersion with self-assembled liposomes was ultra-centrifuged in a Beckman Optima LE-80K ultracentrifuge (Beckman Coulter Inc., Fullerton, CA, USA) using a SW55 rotor at 50,000 rpm (at 4 °C for 1 h). The supernatant containing polymer (PVP) was discarded. The PCS analysis was performed as described by Ingebrigtsen and Brandl, with slight modifications [14]. The specific dilution of the initial dispersion was prepared for the PCS analysis (50× dilution in distilled water). 

### 2.7. Fourier Transform Infrared Spectroscopy 

For verifying drug-polymer interactions of NFs, a Fourier transform infrared (FTIR) spectroscope (IRPrestige-21, Shimadzu Corp., Kyoto, Japan) and Single Reflection ATR crystal (Specac Ltd., Orpington, UK) were used for the HAE, PC and PVP powders, as well as for the electrospun NFs. The analytical range was from 550 cm^−1^ to 4000 cm^−1^ and spectra (*n* = 3) were normalized and scaled.

### 2.8. X-ray Powder Diffraction (XRPD) 

The pure materials were studied by XRPD using the Bruker D8 Advance diffractometer (Karlsruhe, Germany) with a Vario1 focusing primary monochromator (the wavelength of Cu K-alpha 1 radiation = 1.5406 Å), two 2.5 Soller slits and a LynxEye line-detector. For HAE powder, a scanning step of 0.0173°2θ from 5 to 40°2θ and a total counting time of 324 s per step was used. The electrospun NFs were measured with a Goebel mirror (the wavelength of Cu K-alpha radiation = 1.39222Å), two 2.5° Soller slits and a LynxEye line-detector. For NFs, a scanning step of 0.0194°2θ from 5 to 35°2θ and a total counting time of 328 s per step was used. 

### 2.9. Differential Scanning Calorimetry (DSC)

The thermal properties of PVP, PC, HAE and NFs were studied with a PerkinElmer DSC 4000 differential scanning calorimeter (PerkinElmer Ltd., Shelton, CT, USA). The calorimeter was calibrated using indium as a standard. Samples were analyzed under dry nitrogen purge in crimped aluminum pans with 2 pinholes in a lid. The weight of the samples was approximately 3.0 mg. The heating rate was 20 °C/min, and the range for the temperature heating was between 30 °C and 215 °C. Each DSC run was carried out in triplicate. 

### 2.10. In Vitro Drug Release

Drug release studies were conducted in a dialysis bag (molecular weight cut off 10 kDa, Membrane-Cel, Viskase, Inc., Chicago, IL, USA) with 20.0 mg of HAE-loaded NFs re-dispersed in 2.0 mL of phosphate buffered saline (PBS), pH 6.8, as release media. The control sample consisted of 1.0 mL HAE aqueous solution (1 mg/mL) and 1.0 mL of PBS which was placed inside dialysis bags. The dialysis bag was placed in a 50 mL tube containing 20 mL of PBS, pH 6.8. The tube was capped and placed in a dissolution apparatus vessel (Dissolution system 2100, Distek Inc., North Brunswick Township, NJ, USA) with paddles rotating at 100 rpm and maintained at 37 °C. At predetermined time points, 1.0 mL of sample was collected and replaced with a fresh media after sampling. The drug (HAE) content of samples was determined with high-performance liquid chromatography, HPLC (Shimadzu Corporation, Kyoto, Japan) equipped with a Luna C18 25 cm × 4.6 cm, 5 mm C18 analytic column (Phenomenex Inc., Torrance, CA, USA). The HPLC method is described in more detail in the literature [15]. The mobile phase consisted of a solvent A: solvent B mixture (60:40, volume ratio). Solvent A was a 7 mM sodium dodecylsulphate, 25 mM sodium phosphate and 1 mM ammonium acetate solution in a water:acetonitrile mixture (33:67, volume ratio), and solvent B was methanol. The flow rate was 1.0 mL/min and the detector was set at a wavelength of 293 nm.

## 3. Results and Discussion

The grand idea of the present molecular self-assembly strategy is that the active-loaded amphiphilic nanofibrous matrix could serve as a solid template for “inactivate” liposomes under storage, and for the on-demand liposome preparation (“activation”) the present template could be exposed to a small amount of water for spontaneous formation of the liposomes. To date, the formulation of liposomes has encountered a challenge related to the limited physical stability of the final products under storage. The present self-assembled nanofibrous templates introduced by Yu et al. [8] inspired us to utilize such strategy in our study, and it could be a promising approach to resolve this bottleneck in pharmaceutical nanoformulation. To our best knowledge, this area is under researched to date, and would need true attempts to incorporate a therapeutic agent into the amphiphilic NFs to form self-assembled liposomes. 

### 3.1. NMR Spectroscopy Analysis of HAE

Since the plant-origin active agent used in our study (HAE) was isolated from *Zephyranthes ajax* Hort. belonging to the family Amaryllidaceae in Vietnam, the chemical structure of the present alkaloid was verified by means of NMR spectroscopy. The results are summarized in Figure 3 and in the text below. 

Haemanthamine (HAE): Colorless crystal; ^1^H NMR (500 MHz, CDCl_3_): 6.41 (1H, d, *J* = 10.5 Hz, H-1), 6.35 (1H, dd, *J* = 10.5, 5.0 Hz, H-2), 3.86 (1H, m, H-3), 2.11 (1H, ddd, *J* = 13.5, 13.5, 4.0 Hz, H-4), 2.01 (1H, dd, *J* = 13.5, 4.5 Hz, H-4), 4.32 (1H, d, *J* = 16.5 Hz, H-6); 3.68 (1H, d, *J* = 16.5 Hz, H-6), 6.47 (1H, s, H-7), 6.82 (1H, s, H-10), 3.97 (1H, brd, *J* = 4.0 Hz, H-11), 3.35 (1H, dd, *J* = 13.5, 6.5 Hz, H-12), 3.22 (1H, dd, *J* = 13.5, 3.5 Hz, H-12), 5.88 (1H, d, *J* = 4.5 Hz, –OCH_2_O–), 3.35 (1H, s, –OCH_3_); ^13^C NMR (125 MHz, CDCl_3_): 127.4 (C-1), 126.9 (C-2), 72.8 (C-3), 28.3 (C-4), 62.7 (C-5), 63.6 (C-6), 132.0 (C-6a), 106.9 (C-7), 146.2 (C-8),146.5 (C-9), 103.3 (C-10), 135.4 (C-10a), 50.1 (C-10b), 80.2 (C-11), 61.4 (C-12), 100.8 (–OCH_2_O–), 56.7 (–OCH_3_). The present NMR spectroscopy results obtained with HAE agreed well with the previous studies on the corresponding plant-origin alkaloid [16,17]. 

### 3.2. SEM Analysis of Amphiphilic NFs and Templates

Figure 4 shows the representative SEM images of the HAE powder and the individual amphiphilic NFs of HAE as a solid nanofibrous template for self-assembled liposomes. The isolated and milled HAE powder consisted of large irregular particles with a particle size ranging from some tenths of micrometers to several hundred micrometers (Figure 4A). We found that combining HAE and soybean PC with the ES carrier polymer PVP did not impair the performance of an ES process and the formation of NFs. The ES solutions containing HAE, soybean PC and PVP generated continuous elongated NFs with a smooth surface and uniform diameter (Figure 4B,C). The topography of the nanofibrous solid templates exhibited a non-woven and loosely packed platform with randomly oriented individual NFs. The external pore size of the present amphiphilic nanofibrous templates ranged from few micrometers to ten micrometers. Our results are in agreement with those reported by Yu et al. [8] suggesting that the amphiphilic NFs intended for the solid templates for self-assembled liposomes can be fabricated by means of ES.

In our study, the fiber diameter of amphiphilic NFs measured on the SEM micrographs was 392 ± 66 nm (*n* = 100) (Figure 4). The diameter of individual NFs ranged from 197 nm to 534 nm. Yu and coworkers [8] reported that the average fiber diameter of the corresponding amphiphilic NFs (without HAE) ranged from 580 nm to 1250 nm, and the size of the NFs can be tailored by varying the content of PC in the NFs. According to Yu et al. [8], the average diameter of NFs fabricated from pure PVP was 910 ± 110 nm, and there was a significant decrease in the average fiber diameters as the PC content in the NFs was increased (580 ± 90 nm at 33.3% *w*/*w* of PC). However, as the PC content was increased to 50% (*w*/*w*), the average diameter of NFs was significantly increased to 1010 ± 110 nm. The presence of PC, as a zwitterionic surfactant, changes the surface tension and viscosity of the PVP solution, thus affecting the morphology and diameter of the NFs [8].

It is well known that PVP is a good carrier polymer for ES to generate NFs. When adding to PVP solutions, it is evident that PC causes electrostatic hydrophobic interactions with PVP [8]. These molecular-level interactions change the conformation of a PVP chain and PVP–PVP molecular interactions resulting in decreasing entanglements and viscosity [8,18,19]. Taking advantage of the optimal formulation of Yu and coworkers’ study, we carried out the ES of amphiphilic NFs with the same ratio of PC and PVP (1:2 *w*/*w*) for HAE. As shown in Figure 4, the diameter of amphiphilic NFs in our study was much smaller than that obtained by Yu and coworkers (without HAE). Comparing to Yu and coworkers’ study [8], our study involves some differences such as the grade of PVP (in our study K90), organic solvent, polymer concentration and air humidity, which could explain these differences. According to the literature, the changes in the abovementioned variables most likely affect (decrease) the viscosity of the ES solution and modify solvent evaporation, thus contributing a decrease in a fiber diameter [20,21].

### 3.3. Optical Microscopy of Self-Assembled Liposomes

The fate of the self-assembled liposomes in water was monitored by taking optical micrographs at regular intervals after exposing an amphiphilic nanofibrous template to the drop of purified water. As shown in Figure 5, the liposomes were spontaneously formed (self-assembled) in water within few seconds. It was not possible to measure the size of the present self-assembled liposomes of soybean PC and HAE with optical microscopy due to their nanoscale size. The soybean PC releases from the amphiphilic nanofibrous PVP matrix (template) in contact with water and this results in the instant formation of the individual or co-aggregated vesicles (while PVP dissolves in water). Yu et al. [8] reported that PC molecules after releasing from the PVP nanofibrous matrix tend to co-aggregate in water, and the formation of liposomes is dependent on the location of original NFs in the matrix template. These are also in line with our findings. The optical microscopy images obtained in our study suggest that the molecular self-assembly strategy is applicable in the nanoformulation of a plant-origin alkaloid (HAE), and verify the formation of liposomes in water as intended. 

### 3.4. Particle Size Analysis of Self-Assembled Liposomes 

To verify the molecular self-assembly process, the size and size distribution of the soybean PC and HAE containing liposomes was analyzed with PCS. The liposomes exhibited a bi-modal size distribution (Figure 5D). The average diameters of the self-assembled liposomes instantly formed from the hydrated amphiphilic nanofibrous templates of two populations were 63 ± 70 nm (10.3%) and 401 ± 64 nm (89.7%), respectively, with the polydispersity index (PDI) at 0.474. This suggests that in situ formation of liposomes occurred as intended. Yu et al. [8] investigated the hydrodynamic diameter and size distribution of the self-assembled liposomes by static and dynamic light scattering (SDLS), and the average vesicle size and PDI ranged from 64 nm to 369 nm and from 0.182 to 0.299, respectively. Our results with the present drug-loaded self-assembled liposomes are in good agreement with the results reported in the literature with non-drug-containing corresponding liposomes [8].

### 3.5. Physical Solid-State Properties

Physical solid-state analysis was conducted in order to verify potential process-induced solid-state transformations and molecular drug-polymer interactions on the course of the ES of amphiphilic NFs. The XRPD patterns, FTIR spectra and DSC thermograms of HAE (as a powder form) and HAE-loaded amphiphilic electrospun NFs are shown in Figure 6. The XRPD pattern of a pure active agent (HAE) exhibits numerous distinct reflections characteristics to its crystalline nature. The most predominant diffraction peaks of HAE are shown at 12.2, 12.6, 13.8, 16.1, 17.6, 19.6, 20.2, 21.1, 22.6, 23.8 and 27.8°2*θ* (Figure 6A). The XRPD pattern for the HAE-loaded electrospun NFs with two broad amorphous halos indicates most likely an amorphous state of the amphiphilic NFs (Figure 6A). However, the low drug loading in the NFs (approximately 4.5% *w*/*w*) challenges the interpretation of the present XRPD results on the solid state of HAE in the NFs. Since it is generally known that a high-energetic amorphous state fosters the dissolution of the material to water, this finding supports the molecular self-assemble strategy applied for the present nanoformulations of HAE. Our results are also in a good agreement with the findings reported by Yu at al. [8]. They found that the electrospun amphiphilic NFs of soybean PC and PVP (without HAE) are amorphous (XRD), but increasing the amount of PC in the NFs resulted in obvious phase separation (i.e., PC separates out from the PVP matrix template). The amorphous halos observed in the XRPD pattern of amphiphilic NFs are most likely contributed by PVP (Figure 6A).

FTIR vibrational spectroscopy is a powerful tool for detecting the process-induced phase transitions and molecular drug-polymer interactions in the solvent-based manufacturing processes. Figure 6B shows the FTIR spectra of electrospun amphiphilic NFs loaded with HAE and pure materials (HAE, PC, PVP) in a powder form. The FTIR spectrum of HAE shows the distinct peaks in the region of C–H aromatic and aliphatic vibrations ranging from 3051 to 2810 cm^−1^ (Figure 6B). The peaks of the N–H, C–N and C–O groups of HAE were shown at 1475, 1225 and 1053 cm^−1^. The FTIR spectrum of soybean PC shows four bands with corresponding distinct peaks at 2945, 2916, 2847 and 1450 cm^−1^. This finding is also in a good agreement with Yu et al. [8], suggesting the presence of antisymmetric CH_3_ stretching, antisymmetric CH_2_, symmetric CH_2_ and CH_2_ scissoring, respectively. The characteristic band with a single peak at 961 cm^−1^ corresponds to the N^+^ (CH_3_)_3_ stretching vibration (Figure 6B). The two distinct peaks at 1232 and 1056 cm^−1^ are in the region of the antisymmetric and symmetric PO_2_^−^ stretching vibrations [22]. The FTIR spectrum for PVP shows distinct peaks at 2900, 1643 and 1261 cm^−1^, representing antisymmetric CH_3_ stretching, C=O amide stretching and C–N stretching vibration, respectively (Figure 6B). 

Only small changes in the FTIR spectra of the HAE-loaded amphiphilic NFs compared to the spectra of pure materials were found (Figure 6B). The FTIR spectrum for the HAE-loaded NFs exhibited the weak absorbance bands characteristics to HAE and excipients (soybean PC, PVP), thus suggesting the absence of molecular interaction between the three materials used in a solvent-based ES. On the other hand, the concentration of HAE in the NFs is low, thus leading to some challenges to distinguish the characteristic peaks of the active agent. 

DSC was used as a complementary method to verify the drug-excipient compatibility and interactions in the present amphiphilic NFs. The DSC thermogram of HAE showed a single sharp melting endotherm at 205.4 °C (Figure 6C). This is also in a good agreement with the characteristic melting point of HAE (206 °C) reported in the literature [23]. The DSC thermograms for PVP K90 and soybean PC showed a broad endotherm from 50 °C to 120 °C (due to dehydration) and multiple small endothermal events (peaks) ranging from 140 °C to 210 °C, respectively. These multiple fluctuating endothermal peaks for PC can be attributed to the heat-induced movement of polar moieties and the presence of unsaturated bonds, thus resulting in a phase transition from a gel state to a liquid crystal state [24,25]. The DSC thermogram of HAE-loaded amphiphilic NFs showed the characteristic endothermal curve of PVP (ranging from 40 °C to 110 °C), and the absence of the characteristic peak for the melting point of HAE (Figure 6C). Hence, the DSC results suggest that HAE is most likely in an amorphous form in the NFs. The excipients (PVP, PC), however, melt at lower temperatures, and consequently, HAE could dissolve in this melt, thus making it difficult to fully confirm the amorphous state of HAE. Moreover, there are no signs of significant interactions or incompatibilities between HAE and excipients, and the thermograms do not show any signs of chemical decomposition of HAE.

### 3.6. In Vitro Drug Release

The cumulative dissolution profiles of HAE as a powder and loaded in the amphiphilic electrospun NFs are shown in Figure 7. The theoretical amount of HAE in the solid nanofibers sample applied in the dissolution test was 0.89 mg (respective to 100% in the dissolution study). With HAE as a powder, approximately 50% of active ingredient dissolved within the first 2 h and over 80% of HAE dissolved within the next 16 h (after 18 h the dissolution reached the plateau). The release of HAE from the amphiphilic electrospun NFs and self-assembled liposomes occurred in three phases: More than 50% of HAE released within the first 2 h, approximately 80% released within the next 2 h, and the rest of the active agent (100%) released within 30 h (Figure 7). The initial burst release of the drug-loaded amphiphilic nanofibers could be attributed to the surface or near-surface distribution of HAE in the nanofibers and the physical solid state of HAE in the nanofibrous templates. The XRPD results (Figure 6A) suggested that HAE is distributed in an amorphous form in the electrospun nanofibers, thus showing the success of the strategy. In the last phase of the dissolution, a steady-state release pattern was observed obviously due to the drug release from the self-assembled liposomes. Khan and coworkers reported the disintegration and diffusion-controlled mechanism associated with the active release from the nanofibers loaded with oregano essential oil [26]. According to Yu and coworkers, when the amphiphilic NFs are exposed to water, the PC molecules (attached in the PVP chains) are hydrated and concentrated within the nanofibrous network [8]. This results in swelling of the nanofibrous template. The self-assembled liposomes are formed as hydrated PC molecules form co-aggregates, thus entrapping water inside the vesicles. In the final stage, the self-assembled liposomes are released in the dissolution medium [8]. Our dissolution study is only indicative to compare the dissolution properties of HAE and the present amphiphilic nanofibrous templates loaded with the active agent. More research is needed to gain an understanding of the self-assembly of liposomes in the aqueous media and actual release of the active agent from the formed liposomes. 

## 4. Conclusions

The therapeutic use of a plant-origin alkaloid, haemanthamine (HAE), is limited due to the formulation challenges associated with the complex molecular structure and physicochemical properties of the present active agent. We showed that HAE can be formulated to amphiphilic nanofibers (NFs) by solvent-based electrospinning (ES). The amphiphilic NFs provide a solid template for self-assembled liposomes intended to be spontaneously dispersed when the template is exposed to water. The spontaneous formation of self-assembled HAE containing liposomes in water can be proven. The present amphiphilic NFs loaded with HAE are an alternative approach for the formulation of a liposomal drug delivery system and for stabilization of the liposomes of the present alkaloid.

## Figures and Tables

**Figure 1 pharmaceutics-11-00499-f001:**
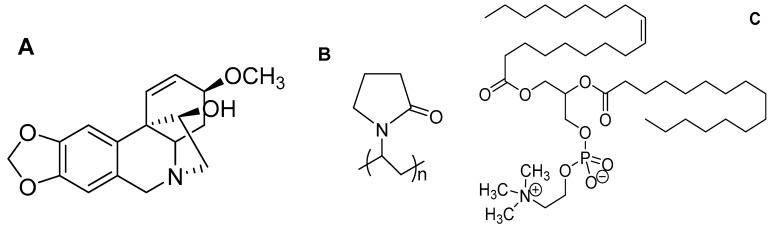
The chemical structure of (**A**) haemanthamine (HAE), (**B**) polyvinylpyrrolidone (PVP) and (**C**) phosphatidylcholine (PC).

**Figure 2 pharmaceutics-11-00499-f002:**
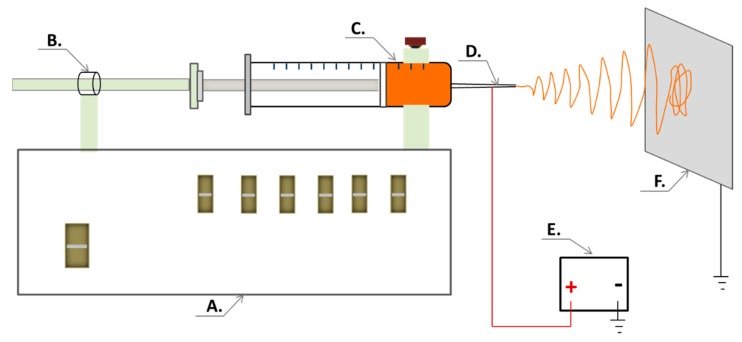
The electrospinning (ES) setup for generating nanofibrous templates for self-assembled liposomes of HAE. Key: (**A**) A robotized ES system; (**B**) programmable syringe pump; (**C**) polymer solution; (**D**) spinneret (a needle system); (**E**) high-voltage power supply; (**F**) collector plate.

**Figure 3 pharmaceutics-11-00499-f003:**
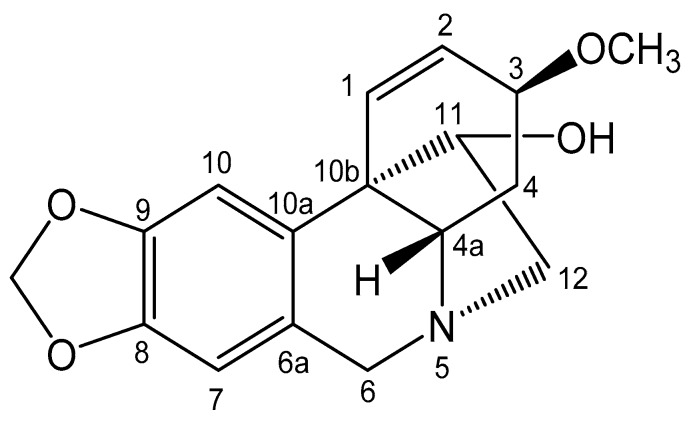
Chemical structure (NMR) of HAE.

**Figure 4 pharmaceutics-11-00499-f004:**
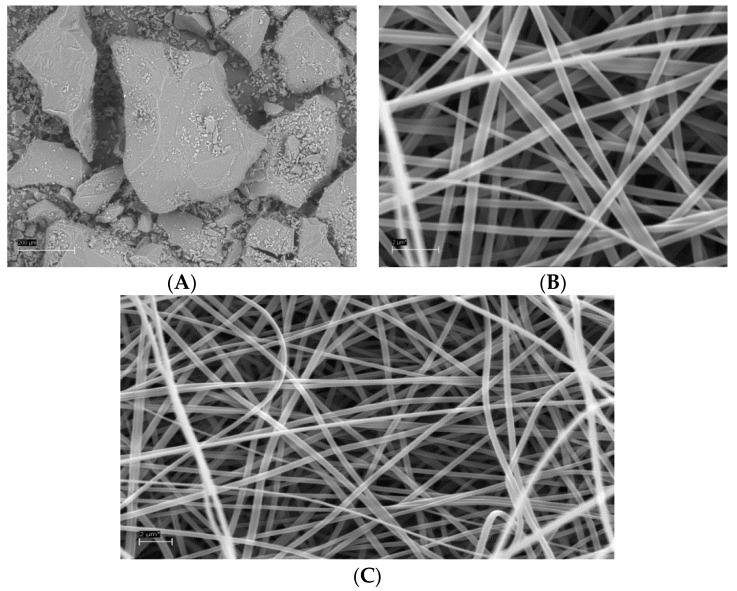
The SEMs of an isolated and milled HAE powder (**A**) and the amphiphilic electrospun NFs (**B**,**C**) used as a solid template for the self-assembled liposomes. Scale bar 200 µm (**A**) and 2.0 µm (**B**,**C**).

**Figure 5 pharmaceutics-11-00499-f005:**
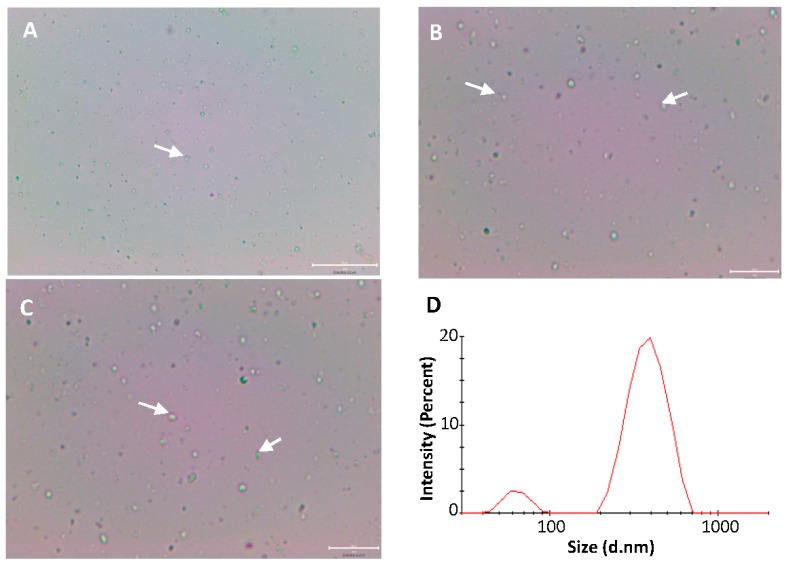
The optical microscopy images (**A**–**C**) and the PCS size and size distribution (**D**) of the self-assembled liposomes in purified water. The liposomes consisting of soybean PC and HAE are spontaneously dispersed from the electrospun amphiphilic nanofibrous template. Due to the limited magnification (50×) of an optical microscope, only the liposomes composed of large vesicles can be seen. Some selected clusters of liposomes are indicated by white arrows. Scale bar 20 µm with 20× (**A**), and 40× (**B**,**C**).

**Figure 6 pharmaceutics-11-00499-f006:**
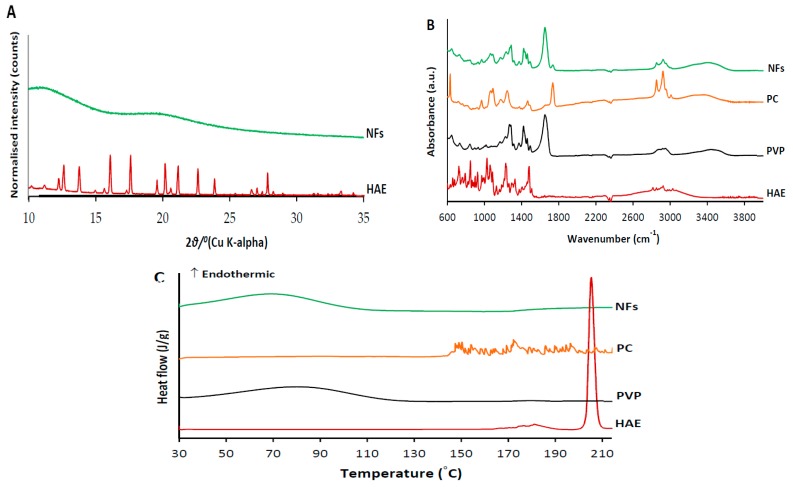
The XRPD patterns (**A**), FTIR spectra (**B**) and DSC thermographs (**C**) of HAE (as a powder form) and HAE-loaded amphiphilic electrospun NFs.

**Figure 7 pharmaceutics-11-00499-f007:**
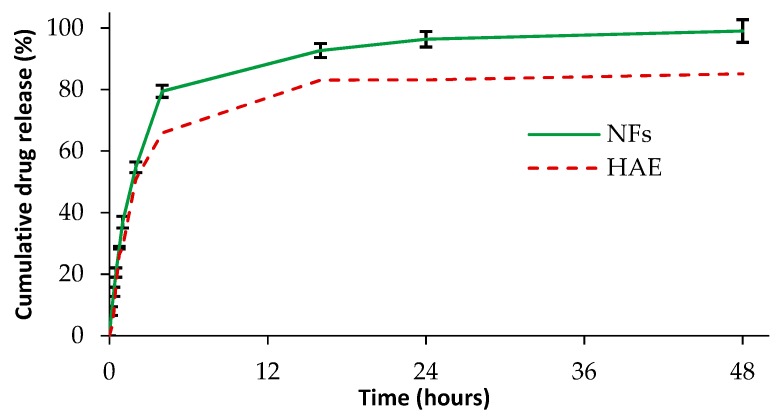
The in vitro dissolution profiles of HAE (as a powder form; a dotted red curve) and HAE-loaded amphiphilic electrospun NFs (a continuous green curve) (*n* = 3).

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
