# Peer review of "Preformulation Study of Electrospun Haemanthamine-Loaded Amphiphilic Nanofibers Intended for a Solid Template for Self-Assembled Liposomes"

_pharmaceutics, 2019, doi:10.3390/pharmaceutics11100499_

Round 1

Reviewer 1 Report

The pharmaceutical quality of the paper is acceptable since all the techniques and characterization methodology are quite consolidated.

As far as novelty I have two observations:

The description of  the preparative method to obtain in situ forming liposomes embedded in nanofibers largely refer to reference 8), therefore the novelty of the proposed formulation should be highlighted in the scope of the paper. The use of a different active (HAE) does not  not per se represent an innovation.

The characterization techniques are quite standard. The only biopharmaceutical improvement is given by dissolution test, but the results are scarcely documented and not brilliant. The relevance to any kind of delivery is not clarified.

The therapeutic goal is not mentioned neither any kind of administration/delivery is envisaged so far, considering the fact that liposomal formulations make sense mainly if administered parenterally.

Given the goal of the journal, the authors should endeavour to provide a comment on the results  obtained and a possible future exploitation.

Author Response

Point 1: The description of the preparative method to obtain in situ forming liposomes embedded in nanofibers largely refer to reference 8), therefore the novelty of the proposed formulation should be highlighted in the scope of the paper. The use of a different active (HAE) does not not per se represent an innovation.

Response 1: As mentioned, our study was originally inspired by the research work of Yu et al. (2011), who introduced polymeric amphiphilic nanofibers as platforms for the self-assembled liposomes. Those amphiphilic nanofibers and templated liposomes introduced and described by Yu et al. did not contain any active agent. Yu and co-workers encouraged scientists to use their innovation as novel biomedical materials and future drug delivery systems. Our study was conducted to further develop the pioneer work of Yu and co-workers, and to extend the use of such amphiphilic nanofibers in the field of Pharmacy. To date, such amphiphilic nanofibers have not been used as drug delivery systems and as a formulation strategy for active pharmaceutical ingredients. To our best knowledge, the study reported here is the first work attempting to load an active agent in the amphiphilic nanofibers to form self-assembled liposomes. The loading of a high-molecule active agent in such nanofibers to maintain the formation of self-assembled liposomes is a paramount challenge. The major difference between our study and the work of Yu et al. was also that we investigated the physical solid-state form transformations and release behavior of active pharmaceutical ingredient loaded in the amphiphilic nanofibers (Yu et al. did not have any active agent in their nanofibers). Furthermore, no formulation studies of a potential plant-origin anticancer drug, haemanthamine (HAE), has been published to date. In summary, the novelty of our work is two-fold: (1) novel drug-loaded amphiphilic nanofibers intended for a solid template for self-assembled liposomes and (2) a novel nanoformulation strategy for a potential new plant-origin anticancer drug, haemanthamine (HAE). The manuscript (“Introduction”) has been revised to better clarify the novelty of our research work (cf. the revised manuscript).

Point 2: The characterization techniques are quite standard. The only biopharmaceutical improvement is given by dissolution test, but the results are scarcely documented and not brilliant. The relevance to any kind of delivery is not clarified.

Response 2: We do agree with your remark. As mentioned, to our best knowledge this is the first study ever using the polymeric amphiphilic nanofibers as solid templates for drug-loaded self-assembled liposomes. The major focus of this work was in the formulation of such active-loaded amphiphilic nanofibers and in the physical solid-state and pharmaceutical characterisation of these nanofibrous templates and self-assembled liposomes. We aimed also to show that the present active-loaded amphiphilic nanofibers are applicable as a novel solid templates for storing “pro-liposomes” prior to self-assembling as the nanofibers are exposed to water. The physical stability is a true challenge and “bottle-neck” with the traditional liposomes. From the formulation point of view, haemanthamine (HAE) is very challenging plant-origin active agent, since it is a large molecule agent, chemically instable (acid-labile), and poorly soluble in water. Therefore, the formulation of high-loaded amphiphilic nanofibers and self-assembled liposomes of such active agent is very demanding. In our study, we investigated the dissolution of HAE loaded in the amphiphilic nanofibers templates and the parallel self-assemble of liposomes. As the drug-loaded amphiphilic nanofibers are dissolved in a phosphate buffered saline, HAE is present in both aqueous solution and in self-assembled liposomes. More research work is needed to gain an understanding of the self-assembly of liposomes in the aqueous media and actual release of an active agent from the formed liposomes. The future research work could also gain our knowledge on the biopharmaceutical improvements and relevance of delivery routes associated with the present nanoformulation. We have revised the manuscript further your suggestion. In the “Introduction” section, the sentence describing the overall goal of the work has been revised (cf. the revised manuscript). In addition, the title of the present study has been revised as proposed by the Academic Editor of the journal. 

Point 3: The therapeutic goal is not mentioned neither any kind of administration/delivery is envisaged so far, considering the fact that liposomal formulations make sense mainly if administered parenterally. Given the goal of the journal, the authors should endeavour to provide a comment on the results obtained and a possible future exploitation.

Response 3: You have indicated an important point here. The plant-origin HAE has been reported to have many prominent bioactivities (potential therapeutic effects), such as neuromuscular transmission, antimalarial, antioxidant, anticonvulsant, antiviral, and anticancer activity. In addition, a number of studies have shown that HAE has a strong cytotoxic activity being presumably effective against certain types of cancer cells (i.e., human melanoma, lung carcinoma, lymphoblast leukemia and breast adenocarcinoma). We are mainly focusing at developing the present novel nanoformulation for anticancer drug therapy and for administering via parental route. It is well known that the sustained-release and/or targeted drug delivery strategies have many advantages in formulating anticancer drugs. These strategies are expected to improve the therapeutic efficiency of an anticancer drug and to reduce the incidence of side effects. With the present active-loaded amphiphilic nanofibers and self-assembled liposomes, further formulation development and optimization are needed to achieve this goal. The manuscript (“Introduction”) has been revised to better clarify the expected therapeutic goal and the route of administration for the readers (cf. the revised manuscript).

Reviewer 2 Report

The authors report on a highly interesting approach to make unstable drugs available for medical treatments. The manuscript is well written and covers an important topic. 

Even though the presence of HAE in the NFs could not be clearly shown from FTIR, the in-vitro dissolution experiments show the success of the strategy. In the discussion of Figure 7, it would be helpful if the authors clarify in the discussion what the respective 100% values relate to.

Minor issues:

p. 2, l.52: the first author of reference 6 is called Pellegrino with family name.

Are Figure 4B and 4C from the same sample type? It may be more helpful to show a comparing SEM for fibres with and without HAE.

p.11, l.329: CH3 - number should be subscript

Author Response

Response to Reviewer 1 Comments

Point 1: Even though the presence of HAE in the NFs could not be clearly shown from FTIR, the in-vitro dissolution experiments show the success of the strategy. In the discussion of Figure 7, it would be helpful if the authors clarify in the discussion what the respective 100% values relate to.

Response 1: We do agree with your remark. The theoretical amount of HAE in the solid nanofibers sample applied in the dissolution test was 0.89 mg (respective to 100% in the dissolution study). To clarify this point, we have added one sentence in the revised manuscript to indicate the respective 100% value. The initial burst release of the drug-loaded amphiphilic nanofibers could be attributed to the surface or near-surface distribution of HAE in the nanofibers and the physical solid state of HAE in the nanofibrous templates. The XRPD results (Figure 6A) suggested that HAE is distributed in an amorphous form in the electrospun nanofibers, thus showing the success of the strategy. In the last phase of the dissolution, a steady-state release pattern was observed obviously due to the drug release from the self-assembled liposomes.

Point 2: p. 2, l.52: the first author of reference 6 is called Pellegrino with family name.

Response 2: This point has been revised as suggested (cf. the revised manuscript).

Point 3: Are Figure 4B and 4C from the same sample type? It may be more helpful to show a comparing SEM for fibres with and without HAE.

Response 3: Figure 4B and 4C are from the same sample type, but they shows the SEM images with different magnification. The two different magnification of SEM images was included in our paper to show the details of the individual HAE-loaded amphiphilic nanofibers (Fig. 4B) and to show the overall structure of the solid nanofiber template (Fig. 4C). For improving the readability, we have enlarged the size of the scale bars in Figure 4. Unfortunately, we do not have any SEM images on the nanofibers without HAE. These polymeric nanofibers without HAE indeed would have been a good reference for the drug-loaded amphiphilic nanofibers. We do apologize for this deficiency in the study protocol. 

Point 4: p.11, l.329: CH3 - number should be subscript

Response 4: The number has been subscripted (CH3) as requested.

Reviewer 3 Report

The work entitled "Development of Electrospun Haemanthamine Loaded Amphiphilic Nanofibers Intended for a Solid Template for Self-Assembled Liposomes" describes an interesting study in a complete and scientifically sound manner. The work is well written and the information clearly presented.The authors did a good job presenting the subject, describing the methodologies and discussing the acquired data. I recommend the publication of this work in the present form. 

Author Response

To Reviewer 2:

We would like to most kindly thank you for your constructive review and for supporting the publication of our manuscript as present form.

Reviewer 4 Report

The present manuscript demonstrates the formulation of Haemanthamine (HAE)-Loaded nanofibers (NFs) via electrospinning. These NFs being prepared from hydrophilic precursors polymers tends to dissolve quickly in water and forms liposomes due to intrinsic property of the phosphatidylcholine. The authors mentioned that the present study has been inspired by the published report (Yu et al. 2011, Reference no.: 8) and the optimised parameters from the reported paper were adapted. The only difference from the previous study was the preparation of the Haemanthamine-Loaded NFs. I am afraid to say that I have severe concerns with the novelty of this work.  

The loading capacity of the NFs has not been explained either in the experimental section or in the result and discussion. Authors directly state that the loading capacity was 4.5% (line no. 304, page 9). This loading capacity is very less and is impractical to be used for the drug delivery applications. Most part of the drug is lost as a burst release and the material is not suitable for sustained drug delivery.

In vitro drug release: Line no. 372, page 12- Author’s states “Our dissolution study is only indicative to compare the dissolution properties of HAE and the present amphiphilic nanofibrous templates loaded with the active agent. More research is needed to gain an understanding of the self-assembly of liposomes in the aqueous media and actual release of an active agent from the formed liposomes.” If the overall objective was to prepare the drug delivery systems, then the authors should have carried out extensive studies to indicate these NFs as potent carriers for drug delivery.

Author Response

Response to Reviewer 3 Comments

Point 1: The present manuscript demonstrates the formulation of Haemanthamine (HAE)-Loaded nanofibers (NFs) via electrospinning. These NFs being prepared from hydrophilic precursors polymers tends to dissolve quickly in water and forms liposomes due to intrinsic property of the phosphatidylcholine. The authors mentioned that the present study has been inspired by the published report (Yu et al. 2011, Reference no.: 8) and the optimised parameters from the reported paper were adapted. The only difference from the previous study was the preparation of the Haemanthamine-Loaded NFs. I am afraid to say that I have severe concerns with the novelty of this work. 

Response 1: As mentioned, our study was originally inspired by the research work of Yu et al. (2011), who introduced polymeric amphiphilic nanofibers as platforms for the self-assembled liposomes. Those amphiphilic nanofibers and templated liposomes did not contain any active agent. Yu and co-workers encouraged scientists to use their innovation as novel biomedical materials and future drug delivery systems. Our study was conducted to further develop the pioneer work of Yu and co-workers, and to extend the use of such amphiphilic nanofibers in the field of Pharmacy. To date, such amphiphilic nanofibers has not been used as drug delivery systems and as a formulation strategy for active pharmaceutical ingredients. To our best knowledge, the study reported here is the first work attempting to load an active agent in the amphiphilic nanofibers to form self-assembled liposomes. The loading of a high-molecule active agent in such nanofibers to maintain the formation of self-assembled liposomes is a paramount challenge. The major difference between our study and the work of Yu et al. was also that we investigated the physical solid-state form transformations and release behavior of active pharmaceutical ingredient loaded in the amphiphilic nanofibers (Yu et al. did not have any active agent in their nanofibers). Furthermore, no formulation studies of a potential plant-origin anticancer drug, haemanthamine (HAE), has been published to date. In summary, the novelty of our work is two-fold: (1) novel drug-loaded amphiphilic nanofibers intended for a solid template for self-assembled liposomes and (2) a novel nanoformulation strategy for a potential new plant-origin anticancer drug, haemanthamine (HAE).

Point 2: The loading capacity of the NFs has not been explained either in the experimental section or in the result and discussion. Authors directly state that the loading capacity was 4.5% (line no. 304, page 9). This loading capacity is very less and is impractical to be used for the drug delivery applications.

Response 2: From the formulation point of view, haemanthamine (HAE) is very challenging plant-origin active agent, since it is a large molecule agent, chemically instable (acid-labile), and poorly soluble in water. Therefore, the formulation of high-loaded amphiphilic nanofibers and self-assembled liposomes of such active agent is very demanding. Actually, the number given in the manuscript (4.5% w/w) does not indicate the loading capacity of the liposomes. We used the phrase “the low drug loading in the NFs (approximately 4.5% w/w)” to indicate the percentage of HAE from the total weight of HAE and excipients in the NFs (nanofibers) (the amount of HAE is 21.0 mg and the amount of excipients PC 0.15 g and PVP 0.30 g in the NFs). The present low percentage illustrates also the challenge to interpret the XRPD results on the solid state of HAE in the NFs. In the future studies, we will make attempts to optimize for example the ratio of active agent and PC to improve the loading capacity of active agent in both the amphiphilic nanofibers and self-assembled liposomes.

Point 3: Most part of the drug is lost as a burst release and the material is not suitable for sustained drug delivery.

Response 3: You have indicated an important point here. The sustained drug delivery plays a crucial role in formulating anticancer drugs. In our study, we investigated the dissolution of HAE loaded in the amphiphilic nanofibers templates and the parallel self-assemble of liposomes. As the drug-loaded amphiphilic nanofibers are dissolved in a phosphate buffered saline, HAE is present in both aqueous solution and in self-assembled liposomes. This in turn results in a burst release observed in the in-vitro dissolution test. HAE is a poorly water-soluble alkaloid of plant origin, and a true challenge from the pharmaceutical formulation point of view. Our results showed that the electrospun amphiphilic nanofibers loaded with HAE could be one alternative to overcome this challenge. It has been shown that the self-deposited liposomes are formed when the pre-weighed nanofibers are added into distilled water (or buffer) and shaken to obtain a homogenous dispersion. The present dispersion is equilibrated for at least 10 min, and un-entrapped portion of active agent in aqueous solution is then removed using the ultracentrifuge. It is evident that the drug release from the liposomal dispersion is slower than that from a nanofibers dispersion. We found, however, that the drug release from the self-assembled liposomes is quite challenging to be verified. As discussed, more research work is needed to gain an understanding of the self-assembly of liposomes in the aqueous media and actual release of an active agent from the formed liposomes.

Point 4: In vitro drug release: Line no. 372, page 12- Author’s states “Our dissolution study is only indicative to compare the dissolution properties of HAE and the present amphiphilic nanofibrous templates loaded with the active agent. More research is needed to gain an understanding of the self-assembly of liposomes in the aqueous media and actual release of an active agent from the formed liposomes.” If the overall objective was to prepare the drug delivery systems, then the authors should have carried out extensive studies to indicate these NFs as potent carriers for drug delivery.

Response 4: We do agree with your remark (reference is also made to previous point). As mentioned, to our best knowledge this is the first study ever using the polymeric amphiphilic nanofibers as solid templates for drug-loaded self-assembled liposomes. The major focus of this work was in the formulation of such active-loaded amphiphilic nanofibers and in the physical solid-state and pharmaceutical characterisation of these nanofibrous templates and self-assembled liposomes. In the “Introduction” section, the sentence describing the overall goal of the work has been revised (cf. the revised manuscript). In addition, the title of the present study has been revised as proposed by the Academic Editor of the journal.  

Round 2

Reviewer 4 Report

Authors have satisfactorily responded to the comments raised. However, I am still interested in knowing the distribution of drug in solution and in the liposomes during NFs dissolution in PBS or water. I highly encourage authors to investigate this distribution in future studies realizing the fact that HAE is a troublesome drug. It is also important to note that the authors should consider performing a systematic study to optimize the drug loading and loading efficiencies.

Author Response

First of all, we would like to thank you for your valuable remarks and constructive criticism. After taking into account of your comments, we feel that our manuscript has been improved a lot.

Your suggestions for the future studies are also very important for us, and this feedback will encourage our research group to continue the work with the present amphiphilic nanofibers and templated liposomes. You are absolutely right in that a systematic further study to optimize the drug loading and dissolution of self-assembled liposomes is needed as a next research step with HAE.

We do believe that the present amphiphilic nanofibers as solid templates for self-assembled liposomes could be a true alternative for the formulation of liposomal drug delivery systems in the future. This is because the stability and manufacturing scaling up issues of pharmaceutical liposomes are still a major “bottleneck” for the commercialization of such products.

Sincerely,

Jyrki Heinämäki